# Uncovering Structural-Functional Coupling Alterations for Alzheimer's Diseases

**Tingting Dan**[1]                                             Tingting_Dan@med.unc.edu

[1] *Department of Psychiatry, University of North Carolina at Chapel Hill, Chapel Hill, NC, USA*

**Guorong Wu**[1,2]                                                  grwu@med.unc.edu

[2] *Department of Computer Science, University of North Carolina at Chapel Hill, NC, USA*

## Abstract

A confluence of neuroscience and clinical studies suggests that disrupted structural connectivity (SC) and functional connectivity (FC) in the brain is an early sign of neurodegenerative diseases. However, current methods lack the neuroscience foundation to understand how these altered coupling mechanisms contribute to cognitive decline. To address this issue, we spotlight a neural oscillation model that characterizes the behavior of neural oscillators coupled via nerve fibers throughout the brain. Tailored a physics-guided graph neural network (GNN), which can predict self-organized functional fluctuations and generate a novel biomarker for early detection of neurodegeneration through altered SC-FC coupling. Our method outperforms conventional coupling methods, providing higher accuracy and revealing the mechanistic role of coupling alterations in disease progression. We evaluate the biomarker using the ADNI dataset for Alzheimer's disease diagnosis.

**Keywords:** Brain structure-functional coupling, Imaging biomarkers, Alzheimer's diseases.

## 1. Introduction

The human brain is a complex system with spontaneous functional fluctuations (Bassett and Sporns, 2017) that can be affected by both normal aging and neuropathology events. Understanding the relationship between structural connectivity (SC) and functional connectivity (FC) (Badhwar et al., 2017) is crucial for identifying effective interventions for diseases such as Alzheimer's (Cummings et al., 2007). Current research examines the statistical association between SC and FC using various approaches (Gu et al., 2021; Park et al., 2008), but lacks a comprehensive understanding of the system-level coupling mechanisms that underlie the emergence of brain functions. To address this, we propose a new approach that leverages established biophysics models to uncover SC-FC coupling mechanisms and generate biomarkers with greater neuroscience insight.

In this regard, we conceptualize the human brain as a complex system where each region is associated with a neural population that exhibits frequency-specific oscillations. Inspired by the success of the Kuramoto model (Kuramoto and Kuramoto, 1984) in modeling coupled synchronization in complex systems, we describe the physical coupling of these oscillatory units via nerve fibers observed in diffusion-weighted MRI images. The resulting phase oscillation process on top of the SC topology generates self-organized fluctuation patterns in the blood-oxygen-level-dependent (BOLD) signal (Fig. 1 top). We propose a novel graph

neural network (GNN) to learn the dynamics of SC-FC coupling from human connectome data and provide insight into the evolving relationship between SC and FC through phase oscillations. Additionally, we propose to use learned system dynamics to yield new SC-FC coupling biomarkers (Fig. 1 bottom). We evaluate these biomarkers using the ADNI dataset (Petersen et al., 2010) and find promising results for recognizing early signs of neurodegeneration, demonstrating potential for future network neuroscience studies.

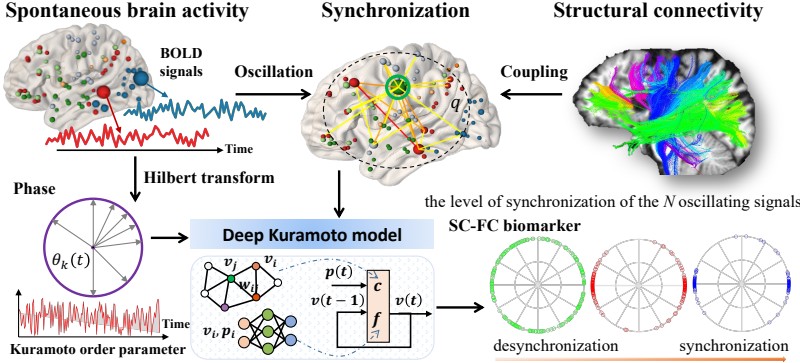

Figure 1: The spatio-temporal learning framework of our proposed deep Kuramoto model.

## 2. Method and Experiment

We model the human brain network $\mathcal{G} = (\Xi, W)$ as a complex system with $N$ brain regions $\Xi = \{\xi_i | i = 1, ..., N\}$ connected by neuronal fibers (i.e., SC) $W = [w_{ij}] \in \mathcal{R}^{N \times N}$, where the neural oscillation status of each region is determined by an intrinsic variable of brain rhythm $v_i(t)$. The synchronized oscillations of multiple brain oscillators give rise to self-organized patterns of functional fluctuations. To test this hypothesis, we propose a deep model that reproduces the topology of the traditional FC matrix $Q = [q_{ij}]_{i,j=1}^N \in \mathcal{R}^{N \times N}$, which is obtained from the BOLD signal $x_i(t)(i = 1, ..., N, t = 1, ..., T)$ of $N$ regions, by using the phase information of neural oscillations. We use the proposed deep Kuramoto model to constrain the synchronization of coupled oscillators.

**Deep Kuramoto Model for SC-FC Coupling Mechanism.** We first propose a general formulation to model a nonlinear dynamical system as:

$$\frac{dv_i}{dt} = f(v_i, \mathcal{H}(x_i)) + \sum_{j \neq i}^N w_{ij} c(v_i, v_j) \tag{1}$$

where the system dynamics is determined by the state variable of brain rhythm $v_i$ on each node. Compared to the classic Kuramoto model (Breakspear et al., 2010), we estimate the natural frequency $\omega_i$ through a non-linear function $f(\cdot)$, which depends on the current state variable $v_i$ and the neural activity proxy $x_i$. We use the Hilbert transform ($\mathcal{H}(\cdot)$) to extract the phase and amplitude information from BOLD signals (Chang and Glover, 2010; Mitra et al., 2015), which has been widely used in functional neuroimaging research.

We formulate the frequency function as $f(v_i, p_i)$, where $p_i = \mathcal{H}(x_i)$ represents the phase information of time course $x_i$. We then introduce a coupling physics function $c(\cdot, \cdot)$ to model the relationship between two state variables $v_i$ and $v_j$, with their coupling strength determined by the structural connectivity $w_{ij}$. The overview of our deep Kuramoto model is shown in Fig. 1. Our input consists of time-invariant coupling information from the SC matrix (top-right) and time-evolving phase information at each node $p_i(t)$ (top-left). In the blue box, our physics-guided deep Kuramoto model captures the dynamics of neural oscillations in a spatio-temporal learning scenario. At each time point $t$, a fully-connected network (FCN) and a GNN (Kipf and Welling, 2016) predict the first and second terms in Eq. 1 for the current state $v_i$ at each node $\xi_i$.

**Novel SC-FC Coupling Biomarkers.** The valuable bi-product of our deep Kuramoto model of neural oscillation is a system-level explanation of how the neuro-system dynamics is associated with phenotypes such as clinical outcomes. In doing so, we introduce the Kuramoto order parameters $\phi_t$ to quantify the synchronization level at time $t$ as $\phi_t = \frac{1}{N} real\{\sum_{i=1}^{N} e^{iv(t)}\}$, where $real(\cdot)$ denotes the real part of the complex number. In complex system areas, $\phi$ is described as the synchronization level, aka. the metastability of the system (Pluchino and Rapisarda, 2006), transiting from complete chaos ($\phi_t = 0$) and fully synchronization ($\phi_t = 1$). In this context, we propose a novel SC-FC coupling biomarker $\mathbf{\Phi} = (\phi_{t_0}, \phi_{t_1}, ..., \phi_{t_T})$ (bottom right corner in Fig. 1) which records the evolution of system metastability underlying the neural activity. *SC-FC-META* uses the $\mathbf{\Phi}$ to conduct the downstream classification tasks, while *SC-FC-Net* is an end-to-end $\mathbf{\Phi}$-training based deep model. The experimental results are shown in Fig. 2, we mainly validate the neuroscience insight (identify brains at risk of AD) based on new SC-FC coupling biomarkers and obtain decent results. This approach holds great promise for other neuroimaging applications.

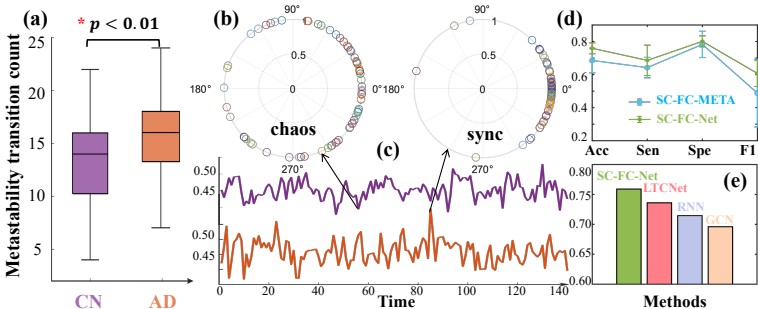

Figure 2: (a) denotes the metastability transition count between CN and AD. (b) Snapshot of node phase visualizations at the chaos and synchronization stages. (c) Global dynamic (order parameter $\phi$) in coupling parameter space. (d) The classification performance ( AD vs. CN) on a shadow approach (SVM, blue) and our *SC-FC-Net* (green) by using our new learning-based SC-FC biomarker. Acc: accuracy, Sen: sensitivity, Sep, specificity, F1: F1-score. (e) The accuracies of diagnosing AD on four methods (LTCNet (Hasani et al., 2021)), GCN (Kipf and Welling, 2016), RNN (Medsker and Jain, 2001).

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
