# OpenReview forum: "Uncovering Structural-Functional Coupling Alterations for Alzheimer's Diseases"
_MIDL.io/2023/Short_Paper_Track — MIDL 2023 Short paper track Poster_

### Official Review · Reviewer_Nx35 · 2023-04-25
**Interesting deep learning approach for solving physical model for neural coupling but details unclear**

**Rating:** 6
**Confidence:** 3

**Review:**

This paper proposes modeling the coupling of structural connectivity (SC) and functional connectivity (FC) using the Kuramoto model, where the terms of the model are predicted by neural networks (a fully connected network for the natural frequency (first) term and GNN for the coupling term). The learned neural dynamics are then used to compute the metastability of the system, which is tested as a potential biomarker for Alzheimer's disease diagnosis through classification of ADNI dataset.

Strengths:
+ Very interesting incorporation of physical model for oscillations of neural activity and coupling structural and functional connectivity
+ Seems to be a very novel approach, using the Kuramoto model but solving using nonlinear neural networks.
+ Usefulness of the learned model dynamics is suggested by derived biomarker for classification of ADNI dataset

Weaknesses:
- Many details of the methods are unclear to me - e.g., I cannot understand if v is a variable we are trying to solve for or if we have this information? How exactly is the model trained - there are 2 neural nets for each of the two terms, but I cannot figure out exactly what is input to the network and what is output and what is trying to be matched. So what is the loss function guiding the training? I thought the model is trying to match the FC matrix q, but q is not mentioned again after its definition.
- Similarly, there are many details absent for the biomarker experiments - what exactly are the classification model architectures, how are they trained, how is data divided?

---

### Official Review · Reviewer_Ch9S · 2023-04-26
**Review of Uncovering Structural-Functional Coupling Alterations for Neurodegenerative Diseases**

**Rating:** 9
**Confidence:** 5

**Review:**

This paper presents a novel method for estimating coupling of structural and functional connectivity. The idea is based on a deep graph neural network (GNN) extension to the Kuramoto model to look at phase oscillation dynamics in the resting-state functional data, coupled to the structural connectivity of the white matter estimated from diffusion MRI.

I think this is a very exciting and novel approach to joint analysis of functional dynamics and structural connectivity. The contribution is more interesting and creative than the typical paper applying existing deep learning methods to medical imaging. The formulation of the deep version of the classical Kuramoto model is well done.

There are just a few suggestions I have for improvement:
1. The results are very brief (necessarily so in a 3-page paper), but one more sentence analyzing what the model says about the differences between AD and controls would be useful. I think the result in Figure 2 (a) could be interpreted and commented on, but I wonder if the multivariate classifier in (d) also has some interesting insight into the differences.
2. The comparison methods in Figure 2 (e) are all cited as machine learning methods (LTCN, GCN, RNN), not specifically papers that were doing AD classification from FC-SC. First, it isn't clear how these models were constructed on this data. Second, it would be useful to compare with previous work (even on the same ADNI data) and the classification rates for AD from FC-SC.